# Zebrafish, an In Vivo Platform to Screen Drugs and Proteins for Biomedical Use

**DOI:** 10.3390/ph14060500

**Published:** 2021-05-24

**Authors:** Hung-Chieh Lee, Cheng-Yung Lin, Huai-Jen Tsai

**Affiliations:** 1Institute of Biomedical Sciences, Mackay Medical College, New Taipei City 25245, Taiwan; d91243003@ntu.edu.tw (H.-C.L.); tonylin0212@gmail.com (C.-Y.L.); 2School of Medicine, Fu Jen Catholic University, New Taipei City 242062, Taiwan; 3Department of Life Science, Fu Jen Catholic University, New Taipei City 242062, Taiwan; 4Institute of Molecular and Cellular Biology, National Taiwan University, Taipei 10617, Taiwan

**Keywords:** zebrafish, high-throughput screening, drug screening, pharmacodynamic

## Abstract

The nearly simultaneous convergence of human genetics and advanced molecular technologies has led to an improved understanding of human diseases. At the same time, the demand for drug screening and gene function identification has also increased, albeit time- and labor-intensive. However, bridging the gap between in vitro evidence from cell lines and in vivo evidence, the lower vertebrate zebrafish possesses many advantages over higher vertebrates, such as low maintenance, high fecundity, light-induced spawning, transparent embryos, short generation interval, rapid embryonic development, fully sequenced genome, and some phenotypes similar to human diseases. Such merits have popularized the zebrafish as a model system for biomedical and pharmaceutical studies, including drug screening. Here, we reviewed the various ways in which zebrafish serve as an in vivo platform to perform drug and protein screening in the fields of rare human diseases, social behavior and cancer studies. Since zebrafish mutations faithfully phenocopy many human disorders, many compounds identified from zebrafish screening systems have advanced to early clinical trials, such as those for Adenoid cystic carcinoma, Dravet syndrome and Diamond–Blackfan anemia. We also reviewed and described how zebrafish are used to carry out environmental pollutant detection and assessment of nanoparticle biosafety and QT prolongation.

## 1. Introduction

### Advantages of Zebrafish for Drug Screening

In recent years, the zebrafish has become one of the most useful model organisms as a rapid means for drug screening in vivo. It has many advantages. First, drug toxicity to embryos during development can be evaluated by simple observation of morphological and developmental defects, as long as the drug is immersed into the medium with embryos. Second, the rapid development of transparent embryos makes it possible to visualize drug efficacy on specific fluorescence- or protein-labeled or dye-stained tissue using an image tracking system. Third, light-induced fecund embryos and low-cost husbandry make large-scale drug screening feasible. Fourth, the genetic maps and sequences of the full diploid genome have been published [1,2,3,4,5], and it was found that the zebrafish genome contains over 70% of human disease-causing conserved gene homologs [5]. Fifth, the zebrafish shares many anatomical and physiological features with humans and has an archetypical vertebrate brain [6]. Sixth, genetic manipulation of zebrafish embryos is simple, but effective, using gain- and loss-of-function approaches by microinjection of mRNAs and antisense oligonucleotide morpholino (MO), respectively, into embryos at one-cell stage. This transient transgenesis is the most convenient approach for drug testing, even though it has only a short-term effect on phenotypes [7]. Seventh, generating a specific germline transmission of zebrafish is fast, economical and effective since transgenic manipulation of zebrafish embryos is relatively easy. Transgenic zebrafish lines are a very reliable means for drug screening. To achieve germline transmission, the transgene flanked with inverted terminal repeats of adeno-associated virus can enhance the ubiquitous expression at F0 generation [8]. Meanwhile, transgenesis can also be facilitated by using the Tol2 transposon derived from medaka [9]. The efficiency of Tol-2-mediated germline transmission could range from 50 to 70% of injected embryos [10]. Nowadays, many zebrafish transgenic lines can act as human disease models. Eighth, gene constructs containing Gal4/UAS [11] and a Tet-On/Off system [12] can facilitate the expression of exogenous DNA under inducible control, while gene constructs containing Cre-loxP [13] can facilitate the deletion of undesired DNA fragments from the DNA construct. Recently, transcription activator-like effector nucleases (TALENs) and clustered regularly interspaced short palindromic repeats (CRISPR) combined with the CRISPR-associated protein 9 (Cas9) system can provide more efficient strategies to perform genomic editing in a specific target gene of the zebrafish recipient [14,15,16].

The above advantages could explain why the zebrafish has been overwhelmingly used as a model organism to study human diseases and has emerged as the perfect animal for high-throughput screening of chemical libraries for potential drugs (Figure 1) [17].

## 2. Embodiments

### 2.1. Using the MO-Knockdown Approach to Silence Genes in the Search for Curative Chemical(s)

In zebrafish, the easiest approach to study loss-of-function is the microinjection of antisense oligonucleotides (MO) to block the translation of target mRNA or stop splicing between exon and intron, resulting in the failure to produce endogenous functional protein [18]. Since defective phenotypes are readily observable in MO-injected transparent embryos, this approach is commonly used to perform drug screening at the early embryonic stage. However, this technique may result in an off-target effect, toxicity and transient and unspecific phenotype [7,19]. Thus, any MO-induced phenotype should go further to validate that phenotype is caused by gene-specific knockdown.

Through MO-knockdown in zebrafish embryos, Fernández-Murray et al. [20] generated a sideroblastic anemia zebrafish model. Interestingly, supplementation with glycine and folate could restore the hemoglobin level in this zebrafish strain, suggesting that this might be a new treatment for SLC25A38 congenital sideroblastic anemia. Additionally, van Karnebeek et al. [21] reported that biallelic deleterious mutations in the *N-Acetylneuraminic acid synthase* (*NANS*) gene are associated with a severe intellectual developmental disorder and skeletal dysplasia. When they used the MO-knockdown approach to inactivate NANS enzymatic activity in zebrafish embryos, it was found that the addition of sialic acid could effectively rescue the defect caused by NANS loss-of-function. Thus, the use of sialic acid is effectively confirmed as a therapeutic strategy for the design of drugs to counteract NANS deficiency.

### 2.2. Using CRISPR/Cas9 Editing to Mutate Gene(s) in the Search for Compounds for Symptomatic and Pain Relief

Several zebrafish human disease models were generated through gene knockout and disease-linked mutations using the CRISPR/Cas9 approach [22,23]. For example, a zebrafish mutant line that carries a *slc39a14* null mutation (*slc39a14^U801^*) generated through CRISPR/Cas9 genome editing was used to elucidate the mechanism underlying childhood-onset Parkinsonism–dystonia [24]. They identified that disodium calcium edetate (Na_2_CaEDTA) could alleviate patients’ symptoms. Meanwhile, using a *scn1Lab* mutant zebrafish, Griffin et al. [25] identified two compounds, Clemizole and Lorcaserin (Belviq^®^; approved by FDA), which exerted antiepileptic activity for Dravet syndrome patients.

Liu et al. [26] reported a zebrafish mutant that contains a defective gene encoding the *low-density lipoprotein receptor* (*ldlr*). The *ldlr*-deficient zebrafish displayed hypercholesterolemia which is aggravated by a short period of high-cholesterol diet. Using this mutant transgenic line, they finally demonstrated that an inhibitor of apolipoprotein B secretion reduced lipid accumulation.

### 2.3. Using Wild-Type Strain to Search for Compounds to Alleviate Clinical Symptoms and Pain

Li et al. [27] used wild-type zebrafish larvae to screen for ginsenosides, i.e., steroid-like saponins with a hypoglycemic effect in a cholesterol diet and identified ginsenoside Rb1 as a potential clinical remedy. Moreover, they treated adult zebrafish with high-dose ginsenoside Rb1 and found a significant decrease of total cholesterol and triglyceride protein levels in the plasma and *LDLR* and *SREBP2* transcripts in the liver.

## 3. Therapeutic Strategy Can Be Found from Phenotypes or Expression Patterns Induced by Chemical Exposure

Urea cycle disorders (UCDs) may cause irreversible brain damage and high mortality. Zielonka et al. [28] employed zebrafish embryos and induced toxicity by lethal NH4^+^ to examine mortality. The results showed that inhibition by ornithine aminotransferase (OAT) could prevent mortality in NH4^+^-exposed zebrafish. This line of evidence might provide clues for the development of new therapeutic strategies potentially applied to patients with UCDs.

Adenoid cystic carcinoma (ACC) is a rare salivary gland-type tumor that arises in exocrine glands, including the salivary, lacrimal, mammary, and bronchial glands. It is now known that MYB-NFIB and MYBL1-NFIB were detected in ~60% of cases of ACC, but with no effective therapy. However, Mandelbaum et al. [29] used a pluripotent zebrafish blastomere culture system to screen 3840 small molecules of bioactive chemicals. They found that the retinoic acid agonist all-trans retinoic acid (ATRA) could effectively decrease the GFP pattern expressed in *c-myb^+^* hematopoietic stem/progenitor cells of zebrafish transgenic line Tg(*c-myb:GFP*) [29]. Interestingly, ATRA was further proven to inhibit tumor growth in patient-derived ACC primagrafts in mice [29]. This treatment is now undergoing a phase II study (ClinicalTrials.gov with the number of NCT03999684).

Aminoglycoside (AG) antibiotics are widely used in clinical practice, but have ototoxicity and nephrotoxicity, resulting in the loss of sensory hair cells from the inner ear [30]. After a large-scale screen of adult zebrafish embryos, Chowdhury et al. [31] found that ORC-13661 provided robust protection of lateral line hair cells exposed to AG antibiotics.

## 4. Screening for Potential Cancer Drugs

Zebrafish has become a popular experimental animal for studies of human cancer [32,33] because (a) zebrafish homologs of human oncogenes and tumor suppressor genes have been identified; (b) the signaling pathways regulating cancer development are conserved [34]; (c) many zebrafish tumors are similar to those of human cancer in histological examination [35]; (d) transparent zebrafish embryos are more accessible for observing the dynamic change of tumor cells; and (e) several transgenic zebrafish lines are available for the in vivo study of angiogenesis modulation [36,37]. Therefore, the zebrafish has emerged as a valuable means as an in vivo evaluation platform for screening anti-angiogenesis drugs from chemical compounds [38], microbial extracts [39] and Chinese herbal extracts [40].

Ethylnitrosourea was the most common mutagen to generate mutant zebrafish. For example, this approach generated zebrafish with a mutation in *tumor suppressor 53* (*tp53^M214K^*) [41], resulting in the development of a wide array of cancer types [42]. Moreover, soaking in heavy metal poison (such as arsenic) was found to induce skin cancer in zebrafish [43]. Additionally, transgenic lines of zebrafish could be applied to establish models simulating cancer development. A melanoma zebrafish model was generated by the expression of the human oncogenic *BRAF^V600E^* gene driven by the zebrafish melanocyte *mitfa* promoter [44]. Several models of liver tumor have been reported using liver-specific expression of transgenic oncogenes, such as *kras*, *xmrk* and *myc* [45,46]. Transgenic lines and zebrafish mutant tumor models could provide abundant resources for mechanistic studies and therapeutic research in human cancer.

Tumor cell transplantation is a relevant method for the assessment of tumor invasiveness. Tumor cells from a donor can be grown in a recipient of the same species (allograft) or another species (xenograft). The allograft on zebrafish by transplantation of melanoma cells from Tg(*mitfa-BRAF^V600E^*);*tp53*^−/−^ transgenic fish to study melanoma pathology and metastatic behavior in adult zebrafish has been reported [47]. Additionally, xenograft is also an important experimental tool for the analysis of normal and malignant cell phenotypes [48]. The xenograft on zebrafish by transplantation of human carcinoma cells can be applied to understand the processes of angiogenesis, tumor cell extravasation, migration, and metastasis, as well as examine the efficacy of drugs for lung, breast, liver and prostate cancer [49,50,51,52,53,54,55,56,57]. Yen et al. [58] developed an optically clear *prkdc^−/−^*, *il2rga^−/−^* zebrafish that lacks adaptive and natural killer immune cells. This mutant zebrafish allows the dynamic visualization of single cells after engrafting different types of human cancers. Moreover, they identified the preclinical efficacy of a combinatorial therapy using Olaparib PARP inhibitor and temozolomide DNA-damaging agent for rhabdomyosarcoma and visualized therapeutic responses. Now, using this same combination therapy to treat Ewing sarcoma and rhabdomyosarcoma has advanced to phase I trials (ClinicalTrials.gov with the number of NCT01858168).

Crossing the blood–brain barrier (BBB) is a major hurdle in the delivery of drugs to treat brain tumors. However, since the zebrafish BBB is functionally and structurally similar to that of mammals, this model can be used to address physiological and pathophysiological situations in vivo [59,60]. More specifically, Yang et al. [61] reported exosome-mediated drug delivery across the BBB of zebrafish. They found that exosome-delivered anticancer drugs could significantly decrease the fluorescence intensity of xenografted cancer cells and tumor growth marker. Furthermore, Zeng et al. [62] established an orthotopic glioblastoma multiforme xenograft model using zebrafish embryos from transgenic line *Tg(flk:eGFP)*. They identified a promising small compound named TNB, which could efficiently cross the zebrafish BBB and inhibit the progression of orthotopic GBM xenografts. These studies affirm the feasibility of crossing zebrafish BBB in combination with a xenografted cancer cell model in a manner that allows studying the transport mechanism, cell uptake, and cytotoxic efficacy of anticancer drugs of human brain cancer cells.

Taken together, this line of evidence suggests that the zebrafish is an excellent alternative animal model for studying the pharmacodynamic profile of drugs used for tumor and cancer treatment.

## 5. Social Behavior and Screening for Potential Drugs

Conventionally, the rodent has served as an animal model to study behavioral disorders. However, they are not suited for scalable, high-throughput and three-dimensional assays. The zebrafish, on the other hand, is also a social animal, but it inhabits three-dimensional space. Moreover, numerous methods allow for the study of zebrafish in their various modes of social behavior. Therefore, the zebrafish has become an attractive animal model for studying behavioral disorders [63].

Autism spectrum disorder (ASD) is a group of heterogeneous neurodevelopmental disorders, characterized by motor, social and cognitive deficits developed during childhood and caused by both genetic and environmental factors. Due to the genetic complexity and pleiotropic nature of ASD, combined with a number of potential environmental causes, it generally confounds the development of potential therapies [64,65]. The rodent model has been used to study most ASD [66,67], but the zebrafish could also be a model for studying the environmental and idiopathic aspects of ASD. In addition to the reasons noted above, the zebrafish model has both cost- and time-effective advantages, as well as the availability of high-throughput screening and quantifiable parameters. For example, Dwivedi et al. [68] developed a zebrafish model to identify novel targets, as well as potential drugs, for ASD. *Contactin-associated protein-like 2* (*CNTNAP2*) is found to be strongly related to autism and epilepsy in consanguineous families [69]. Hoffman et al. [70] used zebrafish *cntnap2*-mutant behavioral profiling as a platform to perform pharmacological screens, resulting in identifying phenotypic suppressors and novel pathways related to ASD. Finally, they found that phytoestrogen biochanin A is specifically able to reverse the mutant behavioral phenotype [70]. Interestingly, although this drug was screened from zebrafish, a lower non-mammalian vertebrate, we noticed that this result could still provide clues to explain why more males are diagnosed with ASD than females, by a 4:1 ratio, namely a positive protective role of estrogens in females [71,72].

Furthermore, the zebrafish was employed to examine the pharmacological effect of drugs on ASD. For example, glutamate NMDA receptor antagonist MK-801 and oxytocin receptor antagonist L-368,899 were found to have effects on zebrafish social interaction and aggression [73,74].

## 6. Rare Disease and Screening for Potential Drugs or Proteins

### 6.1. Introduction

In the absence of a global definition, rare disease may be defined as a one that affects less than 1 in 2000 to 10,000 patients. Now, more than 6000 to 8000 rare diseases have been reported, and 80% of rare diseases have identified genetic origins affecting over 350 million people worldwide. Unfortunately, less than 10% of rare diseases have an approved medical treatment, and many rare diseases are frequently undiagnosed or misdiagnosed. About 75% of rare diseases affect children, and 30% of affected children will die before the age of five [75,76]. Although the advent of novel methodologies, such as next-generation sequencing (NGS), could help identify the primary DNA related to rare diseases quickly, easily, and economically, the resultant information is still ambiguous, owing to unknown causative mutations or unknown gene functions in patients. To solve this problem, we could employ a simple animal model such as zebrafish to understand how a particular mutation results in a phenotype similar to human disease.

### 6.2. Mucopolysaccharidosis (MPS)

Type II MPS (MPS II, or Hunter syndrome) is particularly prevalent in Asia [77]. It is a lysosomal storage disorder characterized by a lack of specific enzymes that break down fats or sugars. More specifically, it is an X-linked disorder resulting from a deficiency in *iduronate 2-sulfatase* (*IDS*), which catalyzes the hydrolysis of the 2-sulphate group of dermatan sulfate and heparan sulfate. An *IDS* gene mutation typically underlies the abnormal accumulation of these two glycosaminoglycans (GAGs), dermatan sulphate (DS) and heparan sulphate (HS), resulting in dysfunction of most organ systems, including the brain, heart, liver, central nervous system, and skeletal system [78,79,80]. Although the manifestations of MPS II are progressive and serious, infants are typically born without any easily identifiable clinical symptoms. Depending on the specific type and severity of the disease, symptoms typically start to present between two and four years of age; thereafter, symptoms progress to the severe phase rapidly [81].

Recently, a zebrafish model was employed to advance the research of this disease. Specifically, knockdown of zebrafish *ids* (*z-ids*) was accomplished by injection of *z-ids*-MO into zebrafish embryos, and the results demonstrated the reduction of z-IDS enzymatic activity and, as expected, the appearance of developmental defects underlying the onset of MPS II pathogenesis [82]. Interestingly, co-injection of recombinant human iduronate-2-sulfatase (Elaprase; idursulfase) combined with *z-ids*-MO could rescue the defects caused by z-*ids* knockdown, suggesting that (1) zebrafish can serve as an in vivo model to study the effect of IDS enzymatic activity on the phenotype and (2) zebrafish can serve as a novel tool to better understand the pathogenesis of lysosomal storage disorder, providing a new method of drug screening for MPS II [82]. Moreover, the zebrafish is helpful in determining the signaling pathway in vivo. For example, Costa et al. [83] reported the role of GAG-Sonic Hedgehog (Shh) axis in the pathogenesis of heart known to occur in MPS II. They demonstrated that loss of z-IDS function led to aberrant heart development and atrioventricular valve malformation in zebrafish embryos. Then, they found that activation of the Shh pathway by the smoothened agonist Purmorphamine (PuA) could rescue the ventricle trabeculation defect in *z-ids* morphants, suggesting that the zebrafish is an alternative tool for MPS II drug screening, potentially contributing to the development of additional novel targeted therapies.

Nevertheless, it is still not fully understood exactly which mutation(s) in the human *ids* gene cause MPS II, even though some mutation sites of *ids* have been identified from clinical samples [84]. Based on the results of new molecular biology techniques, combined with newborn screening [85,86,87], at least 658 mutations have so far been found in the human *ids* gene [Human Gene Mutation Database (HGMD) Professional 2019.1]. However, the fact remains that most *ids* mutations have not been clearly characterized to determine which mutation(s) are sufficient to cause MPS II in humans [84]. To solve this puzzling issue, the zebrafish is also a simple but cost-effective animal model that can serve as an in vivo screening platform. In this particular case, since the preparation of mRNAs for a gain-of-function study is simpler, more effective and economical than that of MO-injection for a loss-of-function study, Lin et al. [88] demonstrated that overexpression of some uncharacterized mutant mRNA, which obviously leads to developmental defects, might also contain a potentially malignant mutation because the enzymatic activity of IDS encoded by this mutant mRNA is inactive. Consequently, the exogenous mutated IDS interrupts the enzymatic activity of endogenous IDS through the dominant negative effect, which, in turn, causes to occur developmental defects. Thus, an approach whereby gain-of-function is reliable and could serve as a reference index to determine which mutated nucleotides of *ids* are tightly related to the occurrence of MPS II, effectively demonstrating the feasibility of function-driven disease gene discovery.

### 6.3. Amyotrophic Lateral Sclerosis (ALS)

ALS is a neurodegenerative disorder that primarily affects the motor neurons (MNs) in the motor cortex, brainstem and spinal cord. ALS starts with progressive limb weakness, swallowing difficulties, and breathing impairment, owing to respiratory muscle weakness, ultimately causing death, usually within two to five years following clinical diagnosis [89]. The disease occurs with an estimated incidence of 1:100,000 and a lifetime risk of 1:400 [90,91]. In ~10% of ALS patients, the disease runs in the family (familial ALS), and at this time, the genes predisposed to ALS have not been identified by a full one third. For the remaining 90% of patients classified as having sporadic ALS, causative mutations have been identified in only 5 to 10% of cases [92]. Thus, the genetic origin of most ALS cases remains to be elucidated.

It has been reported that zebrafish harboring human ALS mutant genes can serve as a model animal to study MN degeneration [93,94]. For instance, overexpression of mutant human SOD 1 (*Cu-Zn superoxide dismutase 1*) in zebrafish led to short motor axons with premature branching accompanied by deficient locomotion [95]. The overexpression of human ALS-related mutations of *TDP-43* (*transactive response to DNA binding protein 43kDa*) resulted in motor behavioral defects and hyperbranched ventral root MNs [96]. The injection of human mutant *FUS* (*fused in sarcoma*) mRNA resulted in motor deficits and ventral root MN axonal abnormalities in embryos [97]. Reduced *c9orf72* (*C9orf72-SMCR8 complex subunit*) function in zebrafish results in motor defects, such as muscle atrophy, MN loss and mortality in early larval and adult stages [98]. Vaccaro et al. [99] demonstrated that the overexpression of mutant human TDP-43 in *C. elegans* and zebrafish models exhibited certain aspects of ALS symptoms, including MN degeneration, axonal deficits, and progressive paralysis. Based on the results shown in these animal models, they discovered that methylene blue is a potent suppressor of mTDP-43 MN toxicity through reduction of the endoplasmic reticulum stress response. They also found that stress inhibitors, such as Salubrinal, Guanabenz and Phenazine, could reduce paralysis, neurodegeneration and oxidative stress in mTDP-43 models [100]. Therefore, ALS-like zebrafish can be used as an in vivo platform for screening therapeutic compounds that might suppress the defective phenotypes.

In both ALS-like models and ALS patients, distal axonopathy has occurred in the pathological scenario of ALS at an early stage. Such pathophysiology exhibits disruption of the neuromuscular junction (NMJ) or reduction of synaptic neurotransmission defects at the NMJ before the appearance of motor deficits and MN loss [101,102,103]. Based on a high-throughput screen of 3765 small-molecule derivatives of Pimozide, Bose et al. [104] identified a small molecule, TRVA242, which significantly rescued the locomotor, motoneuron and NMJ deficiency in ALS models of *C. elegans* with TDP-43 mutant and zebrafish with multiple mutants (TDP-43, SOD1, and C9orf72).

Bruneteau et al. [105] reported that the loss of innervation of NMJ is highly correlated with ectopic expression of NogoA in skeletal muscle. Based on this hypothesis, Lin et al. [106] used two-dimensional PAGE to analyze and compare the total secreted proteins between myoblasts harboring Sol8 control and Sol8-NogoA-overexpressed cells. Surprisingly, they found that the amount of secreted phosphoglycerate kinase 1 (Pgk1) was dramatically reduced in the culture medium of NogoA-overexpressed muscle cells. In contrast, when Pgk1 was transiently knocked out of embryonic zebrafish muscle cells from transgenic line *Tg(mnx:GFP)* via the CRISPR/Cas9 system, defective MNs were observed. Moreover, after intramuscular injection of Pgk1, NMJ integrity and delayed NMJ denervation were observed in adult ALS-like zebrafish (Figure 2A, B). Additionally, muscle contraction of Pgk1-injected leg of ALS-mice exhibited a higher proportion of innervated NMJ and displayed a better locomotion (Figure 2C). This line of evidence suggests that secreted Pgk1 from muscle cells is a determinant of neurite outgrowth of MNs, both in zebrafish and mammals, thus pinpointing a potential protein-drug for treating ALS. In a parallel experiment, Lin et al. [107] generated the zebrafish transgenic line *Tg(Zα:TetON-Rtn4al)* able to conditionally and specifically overexpress Rtn4al/NogoA in the muscle tissue. After induction, *Tg(Zα:TetON-Rtn4al)* embryos displayed NMJ denervation, splitting of myofibrils, body weight loss and less locomotive activity. Based on this evidence, we propose that the zebrafish transgenic line *Tg(Zα:TetON-Rtn4al)* would be a good alternative animal model for studying human neurodegenerative diseases, such as ALS, including drug screening.

## 7. New Compounds Identified from the Zebrafish Screening Platform Have Advanced to Early Clinical Trials

### 7.1. Homeostasis

Hematologic malignancy affects the blood, bone marrow, and lymph nodes. Hematopoietic stem cell (HSC) transplantation has been therapeutically valuable in the treatment of hematologic malignancy over the past 20 years. Using chemical screening, a small molecule named 16,16-dimethyl-PGE_2_ (dmPGE_2_; ProHema) was identified and shown to modulate vertebrate hematopoietic stem cell (HSC) homeostasis in zebrafish [108]. They found that dmPGE_2_-treated zebrafish embryos increased hematopoietic stem cells in the aorta-gonad-mesonephros region. Now, dmPGE_2_ has advanced to phase II clinical trials (ClinicalTrials.gov with the number of NCT00890500).

### 7.2. Diamond-Blackfan Anemia

Diamond-Blackfan anemia (DBA), which is characterized by red blood cell aplasia, is a rare bone marrow failure syndrome. Recently, several zebrafish lines with DBA have been generated and applied to explore novel treatments. Based on this zebrafish screening platform, some highly promising treatments were found and permitted to enter early clinical trials, as follows:(i)Payne et al. [109] found that L-leucine treatment of zebrafish embryos could improve anemia and developmental defects associated with DBA. A pilot phase I/II study of L-leucine in the treatment of patients with transfusion-dependent DBA is in progress (ClinicalTrials.gov with the number of NCT01362595).(ii)Ear et al. [110] reported that treatment with RAP-011(Sotatercept) could dramatically restore the hemoglobin level reduced by ribosomal stress in zebrafish. Sotatercept testing in adults with transfusion-dependent DBA is also in progress (ClinicalTrials.gov with the number of NCT01464164).(iii)Macari et al. [111] employed a mutant zebrafish, *Rps29^−/−^*, which carries a mutated ribosomal gene found in DBA patients, to screen novel compounds. They found several calmodulin (CaM) inhibitors that could successfully rescue the hemoglobin level in the mutant embryos. Subsequently, they studied the effect of the CaM inhibitor trifluoperazine (TFP) in human and murine models. The results supported that TFP treatment may be a very effective therapy for DBA patients. Now, this treatment has been permitted to enter a phase I/II clinical trial (ClinicalTrials.gov with the number of NCT03966053).

### 7.3. Dravet Syndrome

Dravet syndrome (DS) is a catastrophic pediatric epilepsy with severe intellectual disability, impaired social development and persistent drug-resistant seizures. It has been reported that 80% of patients have a mutation at the *Nav*1.1 (SCN1A) gene encoding a voltage-gated sodium channel. Therefore, Baraban et al. [112] generated a transgenic zebrafish line containing a *Na_v_*1.1 (*scn1Lab*) mutant and employed this line to screen 320 compounds. Based on phenotypic observation, they identified a U.S. Food and Drug Administration-approved compound, named Clemizole, which is able to inhibit convulsive behavior and electrographic seizures. Again, this compound has been permitted to enter a phase II clinical trial (ClinicalTrials.gov with the number of NCT04462770).

### 7.4. Congestive Heart Failure

Tsai’s lab is the first to generate a transgenic zebrafish line with myocardium-specific GFP expression, in which a DNA fragment containing *GFP* cDNA driven by zebrafish *cardiac myosin light chain 2* (*cmlc2*, now renamed as *myl 7*) is harbored [113]. Furthermore, his lab developed the first myocardium-specific Tet-On system in zebrafish, in which antisense RNA of the cardiac troponin C gene and GFP mRNA were generated conditionally. Upon exposure to doxycycline inducer, adult zebrafish from this line exhibited atrial and ventricular asynchrony, greater cardiac chamber, slower heart rate and lower ventricular ejection fraction (Figure 3) [114]. These results suggest zebrafish as a new, simple animal model with phenotypes simulating dilated cardiomyopathy owing to incomplete atrio-ventricular block. Therefore, this transgenic line would be a good alternative to study human heart failure and perform in vivo large-scale screening of new calcium sensitizer drugs or natural extracts with therapeutic potential for the treatment of contractile dysfunction in congestive heart failure.

## 8. Pollutant and Ecotoxic Study and Screening for Potential Drugs

Environmental pollutants comprise serious threats to living organisms and fragile ecosystems. Hence, monitoring environmental changes constantly is a crucial element to predict any possible contaminants. Detecting environmental pollution at the early stages is a prerequisite for preventing further negative impacts on ecosystems. In the last 20 years, the convergence of molecular toxicology, genomics, high-throughput screening and systems biology aligned with other technological advances has pointed researchers toward the use of the zebrafish as a toxicological model. Indeed, several transgenic lines of zebrafish have been reported as a useful means to detect environmental toxicants and mutagens [115]. Unlike *Tg(cyp1a:gfp)*, which was specifically used to detect Dioxins/Dioxin-like compounds and polycyclic aromatic hydrocarbons [116], transgenic line *huORFZ* [117,118] was responsive to a mixture of pollutants, including both known and uncharacterized compounds, since this line contains a GFP reporter gene controlled by an upstream open reading frame (uORF) of the human *CHOP* mRNA cassette. It displayed no detectable leakage under the normal condition, while it responded to environmental stresses, resulting in translating GFP reporter, with unique GFP expression patterns responsive to various stresses (Figure 4). Importantly, they found that GFP displays when *huORFZ* embryos are exposed to pollutants when levels exceed the WHO standard [118]. Therefore, the *huORFZ* transgenic line serves as a reliable bioindicator for aquatic environment monitoring and an organismal tool for screening toxicant-free material. Moreover, it can be used to investigate the mechanism of action of environmental toxins and their related diseases.

Amazonian wildfires release a large number of gases and pollutants with concomitant serious impacts on human health, such as pulmonary disease, nerve disorders, atherosclerosis, and even cancer from long-term inhalation [119,120]. The thermal degradation of lignin by-products is mainly pointed out as the sole emissions from these fires. Pyrolysis of lignin results in the formation of fuel-specific methoxyphenols. Moreover, complex migration and transformation processes of methoxyphenols in the troposphere can result in the formation of new airborne pollutants and increase the secondary organic aerosol yield. Babić et al. [121] used a zebrafish wild-type strain as a high-throughput screening platform to evaluate the toxicity of methoxyphenols formed during lignin pyrolysis, including guaiacol, catechol, and their nitrated forms: 4,6-dinitroguaiacol, 5-nitroguaiacol and 4-nitrocatechol. The whole-organism bioassay integrated with molecular modeling revealed that methoxyphenols inhibit tyrosinase, lipoxygenase, and carbonic anhydrase, which, in turn, alters embryonic development. This line of evidence confirmed the harmful effects of lignin degradation products and their intermediates on aquatic organisms.

## 9. Finding the Effect of Prescribed Medicines on Developing Embryos

The zebrafish embryo is an excellent means to study whether pre- or post-clinical medicines can affect developing embryos. For example, when zebrafish embryos were used to screen some class D drugs, we found that Amiodarone could cause the regurgitation of blood in the heart. Amiodarone is a type III anti-arrhythmic drug commonly administered to patients with cardiac arrhythmias, such as atrial fibrillation, severe congestive heart failure or after acute myocardial infarction. Chen et al. [122] used the zebrafish transgenic line *Tg(cmlc2:HcRFP)* generated by Tsai’s lab to demonstrate that Amiodarone could impair valve formation in the developing heart of zebrafish embryo since Amiodarone induces ectopic overexpression of *s-versican* in myocardium, resulting in a reduced amount of p-EFGR in the endocardium, which in turn, increases *cdh5*, thereby hindering cell migration and, hence, the formation of valves at the AV canal.

Meanwhile, Hoppstädter et al. [123] used zebrafish wild-type strain to demonstrate the adverse effects of preclinical statins, which are the first-line drugs prescribed for hyperlipidemias and the prevention of cardiovascular disease.

## 10. Zebrafish Serves an In Vivo Platform to Study the Properties of Nanoparticles

Nanomaterials are overwhelmingly used in the health sciences. The zebrafish has mainly been used as an in vivo model to evaluate nanotoxicity, nanomedicine and nanosafety [124,125,126,127]. For example, Javed et al. [128] used zebrafish to demonstrate the utility of casein-coated gold nanoparticles (AuNPs) in sequestering intracerebral amyloid beta (Aβ42) and its elicited toxicity in a nonspecific chaperone-like manner, resulting in recovering the mobility and cognitive function of adult zebrafish exposed to Aβ42.

However, the most beneficial use of the zebrafish model for in vivo study of nanoparticles (NPs) properties involves the application of multicolor fluorescence imaging. This system employs a combination of reporter proteins and fluorescence tracers that can be observed in a basic arrangement at high spatiotemporal resolution. Hayashi et al. [129] used transgenic zebrafish embryos for intravital real-time and ultrastructural imaging of nanomaterials. They used intravital confocal, electron and correlative light-electron microscopy to directly visualize NPs sequestration by macrophages and scavenger endothelial cells in real time at ultrastructural resolution.

The chorion of zebrafish embryos contains pore channels 300 nm^−1^ μm in diameter, acting to shield embryos from the environment [124]. Interestingly, recent reports have shown that it can be applied to study the biodistribution of NPs. For instance, Chen et al. [130] examined how the zebrafish chorion affects the transport of silver nanoparticles (AgNPs). They found that larger-sized (50 nm) AgNPs could be transported through pores of the chorion and distribute into embryos more efficiently than smaller-sized (10 nm) AgNPs.

## 11. Zebrafish Serves as an Excellent Alternative to Screen the QT Prolongation of Drugs

Long QT syndrome (LQTS) is characterized by prolongation of heart rate-corrected QT interval and dysmorphic T-waves on surface electrocardiogram recordings. LQTS may increase the risk of fainting, drowning, seizures, or sudden death owing to torsade de pointes [131,132,133]. Inherited LQTS has been linked to mutations in several cardiac ion channel genes, including *KCNQ1*, *KCNH2*, and *SCN5A* [133]. The *KCNH2* gene encodes the α-subunit Kv11.1 (hERG), which is important for the physiological suppression of early afterdepolarizations (EADs) and triggering activity. Torsadogenicity, usually associated with blockage of hERG channels, occurred in approved drugs subsequently withdrawn from the market. Therefore, since 2005, mandatory guidelines in Europe, the USA and Japan have required the examination of all drug candidates for their hERG liabilities before filing an investigational new drug (IND) application [134,135].

Interestingly, although zebrafish contain one atrium and one ventricle, recent studies have demonstrated that zebrafish provide a viable in vivo model of cardiac electrophysiology because the heart rate, action potential characteristics and ECG morphology of zebrafish are similar to those of humans, compared to mice, rats and rabbits [136,137,138,139]. For example, the action potential duration of mice is extremely short as a result of large repolarization potassium current, resulting in no visible T wave or QT internal in ECG [140,141,142]. Therefore, the zebrafish has become a useful animal to estimate QT prolongation caused by novel drug candidates. For example, Hull et al. [143] reported that the adult zebrafish heart can be used as a valuable, ex vivo whole heart model for the pharmacological investigation of both hKCNH2a channel blockers and activators. Additionally, zebrafish are used to examine known LQTS-causing mutations, screen effects of mutations found in humans, and discover and elucidate effects of novel mutations that affect cardiac repolarization [144,145,146,147].

## 12. The Limitations and Challenges of Zebrafish Model for Drug Screening

Compared to the mouse model, we have suggested many advantages of using zebrafish for drug screening. The zebrafish can serve as a good intermediate alternative to bridge the big gap between the high-throughput screening of chemical libraries for potential drugs in cellular models lacking physiological context and the low-throughput and costly mammalian models that are closer to human physiology (Table 1). Still, some limitations and challenges should be acknowledged. First, unlike mammals, zebrafish lack heart septation, lung, mammary gland, prostate gland and limbs [32]. Thus, screening for drugs that might affect those organs would be impossible. Additionally, zebrafish have no placenta, causing zebrafish embryos to be directly exposed to the environment. Second, in-water dosing may yield unforeseen consequences compared to the typical mammalian routes. Zebrafish are immersed and/or swimming in the treated solution. Third, zebrafish are poikilothermic and are usually maintained below 30 °C, while mammals have, through evolutionary metabolism, adapted to 37 °C. Fourth, although almost all human genes can be found in the zebrafish, approximately 20% of human genes have two orthologues in zebrafish [5,148]. If two orthologues are present, they often show different expression patterns. These duplicated genes will significantly complicate the generation of knockout/in strains using either forward or reverse genetics approaches. Finally, zebrafish and mammalian models have totally different physiology. This means that the translation of pharmacological findings to higher vertebrates is very limited without dose conversion between zebrafish and human based on an internal exposure–response relationship [149,150]. Finally, the basic pharmacokinetic (PK) processes of zebrafish are limited, such as absorption, distribution, and clearance, which drive the exposure of drug and its metabolites. Nevertheless, a few studies suggest quantitative similarities in metabolite formation between zebrafish larvae and humans [151,152,153]. Furthermore, Kantae et al. [154] developed a new methodology to characterize the drug elimination processes in three-day-old zebrafish larvae, proving for the first time the feasibility of studies on drug pharmacokinetic and metabolism in zebrafish. However, more evidence is needed. Although zebrafish have been used as a model animal for the last two to three decades, more time is still needed to accumulate evidence for this issue.

## 13. Conclusions

Despite the numerous physiological differences, such as blood/plasma composition, host immune response and temperature consistency, between zebrafish and humans, the zebrafish is still a powerful and economical in vivo animal model for drug screening and assessing the effect of drugs on embryonic development. Moreover, taking advantage of the improved efficiency and availability of genome editing to produce genetically modified and mutant zebrafish, more and more human genetic diseases could be generated in the zebrafish model. Before moving on to more laborious pharmacokinetic studies and more costly mammalian models, the zebrafish is a good intermediate alternative as an in vivo platform to screen potential drugs on a large scale. This approach would not only contribute significantly to the literature, but also facilitate the implementation of innovative, comprehensive and cost-effective testing strategies.

## Figures and Tables

**Figure 1 pharmaceuticals-14-00500-f001:**
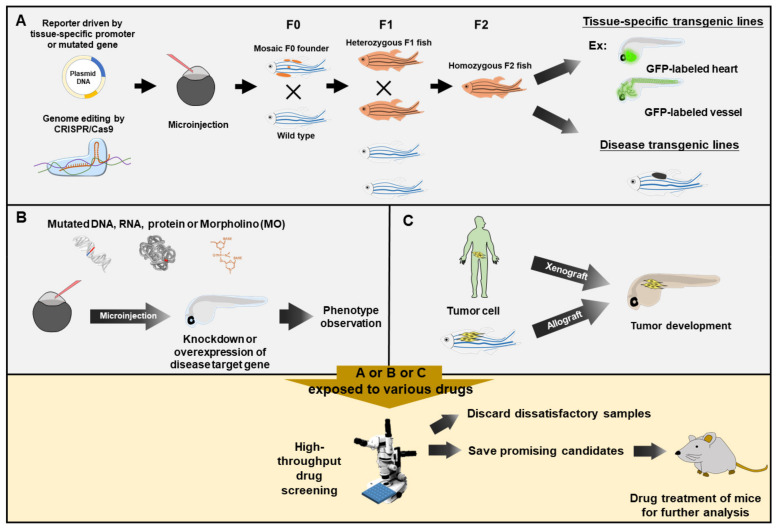
Zebrafish served as a rapid means for phenotype-based high-throughput drug screening. Zebrafish embryos exposed to various drugs for drug screening would be obtained from three approaches. (**A**) Embryos from tissue-specific transgenic lines and disease-like transgenic lines. The DNA constructs, as indicated, are microinjected into one-cell fertilized eggs, followed by selection and breeding to heterozygous or homozygous transgenic strains. (**B**) Embryos with knockdown or overexpression of a specific gene. The mutated DNA, mRNA, protein or antisense oligonucleotide Morpholino (MO) is injected into the zebrafish embryos for knockdown or overexpression of disease target gene, followed by transient phenotypic observation. (**C**) Embryos after transplantation. Cancer cells from human (Xenograft) or zebrafish (allograft) are transplanted into zebrafish embryos, resulting in tumor development. Zebrafish embryos obtained from (**A)**, (**B**) or (**C**) are employed to perform high-throughput drug screening via exposure to various tested drugs by using automated quantitative analysis and phenotypic scoring in multiple plates. After preliminary high-throughput rapidly screening, researchers could rule out most negative candidates and only save the most promising candidates for further validation on the mammalian system.

**Figure 2 pharmaceuticals-14-00500-f002:**
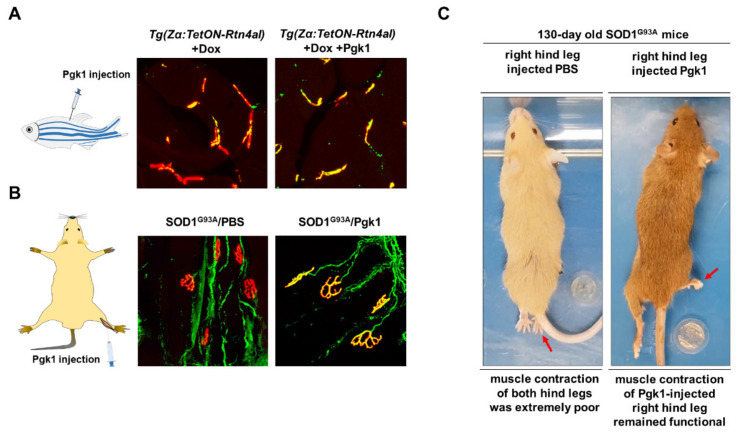
Intramuscular injection of Pgk1 could reduce the NMJ denervation occurring in the ALS animal models of zebrafish and mouse. (**A**) ALS-like adult zebrafish model. Left: diagram illustrates the muscle injection of Pgk1 into adult zebrafish; middle: after Doxycycline (Dox) was treated for one week, a few number of axonal motor neurons (MNs; labeled with green fluorescence signal) co-localized with motor end plate (labeled with red fluorescent signal) presented in yellow fluorescent signal was observed in the ALS-like zebrafish strain *Tg(Zα:TetON-Rtn4al)*, indicating that NMJ denervation, a pathogenesis indicator at early stage of ALS, seriously occurred in the Rtn4al/NogoA-overexpressed zebrafish; right: Pgk1 was injected into the muscle of Dox-treated adult zebrafish, and the number of axonalMNs (labeled with green fluorescence signal) co-localized with motor end plate (labeled with red fluorescent signal) presented in yellow fluorescent signal was increased, indicating extracellular injection of Pgk1 could reduce the denervation and maintain the complete NMJ structure in the Rtn4al/NogoA-overexpressed zebrafish. (**B**) ALS (SOD1^G93A^) mice model. Left: diagram illustrates that the solution was injected into the gastrocnemius of right hind leg of ALS mice; middle: phosphate buffer saline (PBS) served as control was injected into the gastrocnemius of right hind leg of ALS mice, and the denervation of axonal MNs was examined. A few numbers of axonal MNs (labeled with green fluorescence signal) co-localized with motor end plate (labeled with red fluorescent signal) presented in yellow fluorescent signal was observed in the 75-day-old ALS mice; right: Pgk1 was injected into the gastrocnemius of right hind leg of ALS mice at 60 days old, and the denervation of axonal MNs was examined at 75 days old. The number of NMJ with complete structure presented a yellow fluorescent signal was increased compared to that of control group. (**C**) The first shot of Pgk1 was injected into the gastrocnemius of right hind leg of SOD1^G93A^ ALS-mice at 60 days old, followed by the other shot for every 15 days until 120s days old. The treated ALS-mice were observed at 130 days old. Left: both hind legs of ALS-mice injected with PBS (control) were completely paralyzed. Right: the right hind leg injected with Pgk1 could keep contraction and movable, while the left hind leg was paralyzed. Figure modified from [106].

**Figure 3 pharmaceuticals-14-00500-f003:**
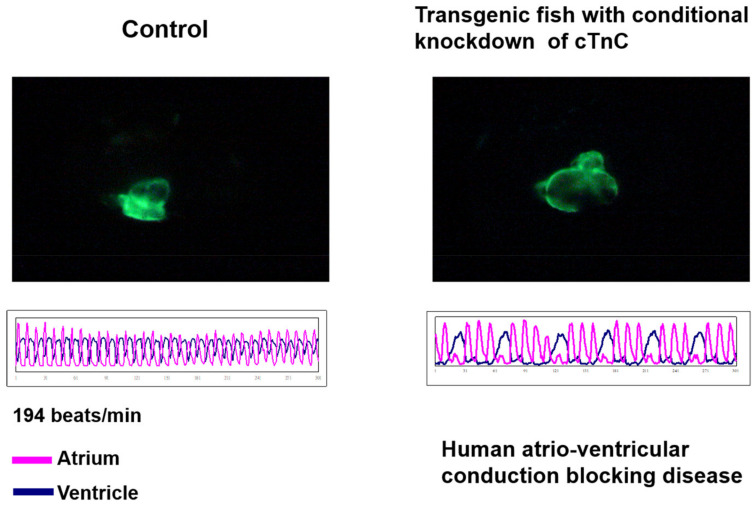
Dilated Cardiomyopathy could be conditionally induced to occur in the transgenic zebrafish. Zebrafish transgenic line *Tg(cmlc2:tetON-cTnc-antisence)* was generated by Tsai’s lab, in which the antisense RNA strand of cardiac Troponin C (cTnC) gene and myocardium-specific GFP mRNA were conditionally induced to synthesize simultaneously in the myocardial cells of the heart; due to this, both directional transcriptions were driven by the promoter of *cardiac myosin light chain 2 gene* (*cmlc2*) under the Tet-On control system [114]. The heart shape (upper panel of left figures) and heartbeat (lower panel of left figures) were apparent normally in the embryos derived from this transgenic line without treatment of Doxycline (DOX) (control). However, when these embryos were treated with DOX, the dilated cardiomyopathy of heart shape occurred (upper panel of right figures), which was able to be observed under confocal microscope, and arrhythmic heart-beating was also detected (lower panel of right figures). These symptoms shown on zebrafish caused by knocking down of cTnC were similar to those of human incomplete atrio-ventricular conduction blocking disease.

**Figure 4 pharmaceuticals-14-00500-f004:**
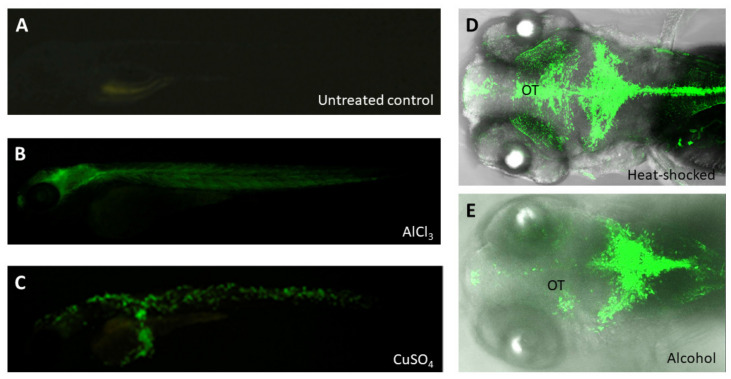
The GFP reporter was expressed in the embryos from zebrafish transgenic line *huORFZ* exposed to various stresses. (**A**) The *huORFZ* embryos without any treatment displayed no GFP signal during 96 hpf served as negative control. (**B**,**C**) The heavy metal-containing chemicals, including AlCl_3_ and CuSO_4_, were used individually to treat *huORFZ* embryos at 72 hpf and observed at 96 hpf [118]. (**B**) For AlCl_3_-treated *huORFZ* embryos, GFP signals were observed in the brain and muscles. (**C**) For CuSO_4_-treated *huORFZ* embryos, GFP signal was observed in the skin cells. (**D**,**E**) The *huORFZ* embryos at 72 hpf were individually treated with heat shock or 1.5% alcohol and then observed the GFP signal during 96 hpf. Although GFP was induced to express by both stresses at the CNS, the expression patterns of them were distinct. (**E**) Alcohol-treated *huORFZ* embryos displayed weak GFP signal at the optic tectum (OT).

**Table 1 pharmaceuticals-14-00500-t001:** List of examples using zebrafish as a platform for drug screening.

Disease	Mutant/Transgenic Line	Size of Screened Library	Identified Drug	Ref.
Adenoid cystic carcinoma	Tg(*c-myb:GFP*)	3840	ATRA	[29]
Amyotrophic lateral sclerosis	UAS:GR transgenic line crossed with Tg(*Hb9*)	3765	TRVA242	[104]
Aminoglycoside antibiotics	Wild-type adult zebrafish	99	ORC-13661	[31]
Dravet syndrome	*scn1Lab* mutant zebrafish	370	Lorcaserin	[25]
Dravet syndrome	*scn1Lab* mutant zebrafish	320	Clemizole	[112]
Hematologic malignancy	Wild-type embryos	2480	ProHema	[108]

## Data Availability

Not applicable.

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
