# Peer review of "Zebrafish, an In Vivo Platform to Screen Drugs and Proteins for Biomedical Use"

_pharmaceuticals, 2021, doi:10.3390/ph14060500_

Round 1

Reviewer 1 Report

Lee et al. provide a review summarizing the literature around the options of zebrafish screens for identifying compounds in biomedical applications. The style of writing is generally very well and the article is pleasant to read. The topics of the review cover a wide field from cancer, heart disease, rare diseases, social behaviours, newly identified drugs progressed to clinical trials, environmental pollutants and nanoparticles. A short introduction into zebrafish biology and then into each topic/disease is given followed by highlights and new advances found by screens in the respective field. The article argues the case for the usefulness of zebrafish screens.

There are several points that I would like to make:

(A) As the review deals with a vast array of different diseases it seems to me that the review is actually addressed to researchers working on the respective diseases who are potentially interested in how such zebrafish screens are performed or are planning to do such a screen themselves. In that case it would be actually helpful to include a schemata/figure on the experimental strategy and setup (either general or an example). Besides that, most of the readers will not be in the zebrafish field and therefore visualizing the setup including a timeline of zebrafish development and when/how drugs can be administered would improve the review.

(B) A table summarizing the literature (or a schemata) on the different possible applications of zebrafish screens (e.g which diseases/which mutant or MO /size of screened library/identified drugs ….) would support the review.

  1. Some minor points: line 38: the full diploid genome; line 73: the meaning of this sentence is unclear: “Also the transparency…”; line 212 (Ref format); line 459 word missing “it can be used”; line 501: one nanotoxicity too many, “evaluate” ; line 514: 0.7mm? what´s the right unit; ?; line 517-19: Is it AgNPs or AuNPs -abbreviation for AgNPs is not spelled out?

One further comment: I am not aware if it is valid to characterize a whole animal such as zebrafish with the term biomaterial.

Author Response

Reviewer 1

Lee et al. provide a review summarizing the literature around the options of zebrafish screens for identifying compounds in biomedical applications. The style of writing is generally very well and the article is pleasant to read. The topics of the review cover a wide field from cancer, heart disease, rare diseases, social behaviours, newly identified drugs progressed to clinical trials, environmental pollutants and nanoparticles. A short introduction into zebrafish biology and then into each topic/disease is given followed by highlights and new advances found by screens in the respective field. The article argues the case for the usefulness of zebrafish screens.

There are several points that I would like to make:

  • As the review deals with a vast array of different diseases it seems to me that the review is actually addressed to researchers working on the respective diseases who are potentially interested in how such zebrafish screens are performed or are planning to do such a screen themselves. In that case it would be actually helpful to include a schemata/figure on the experimental strategy and setup (either general or an example). Besides that, most of the readers will not be in the zebrafish field and therefore visualizing the setup including a timeline of zebrafish development and when/how drugs can be administered would improve the review.

Authors’ response to #A:

Thank you for your suggestion. In this revised manuscript, we added one figure (Fig.1) to illustrate how such zebrafish screens are performed as follows. Please see line 81 in the revised manuscript.

  • A table summarizing the literature (or a schemata) on the different possible applications of zebrafish screens (e.g which diseases/which mutant or MO /size of screened library/identified drugs ….) would support the review.

Authors’ response to #B:

Thank you for your suggestion. In this revised manuscript, we added one table (Table 1) to summarize the literature on the different possible applications of zebrafish screens. Please see line 612 in the revised manuscript.

  1. Some minor points: a. line 38: the full diploid genome; b. line 73: the meaning of this sentence is unclear: “Also the transparency…” (未改); line 212 (Ref format); c. line 459 word missing “it can be used”; d line 501: one nanotoxicity too many, “evaluate”;e line 514: 0.7mm? what´s the right unit; ?;f line 517-19: Is it AgNPs or AuNPs -abbreviation for AgNPs is not spelled out?

Authors’ response to #1:

Thank you for pointing out these typos. We corrected them.

  1. Please see lines 49-50 in the revised manuscript.
  2. Please see lines 91-93 in the revised manuscript.
  3. Please see line 507 in the revised manuscript.
  4. Please see line 549 in the revised manuscript.
  5. Please see lines 565-566 in the revised manuscript.
  6. Thank you. We add the information to describe the abbreviation of AgNPs. Please see line 569 in the revised manuscript.

  1. One further comment: I am not aware if it is valid to characterize a whole animal such as zebrafish with the term biomaterial.

Authors’ response to #2:

As your suggestion, we replaced “biomaterial” by “means” in the whole manuscript as follows:

“In recent years, the zebrafish has become one of the most useful model organisms as a rapid means for drug screening in vivo.

Transgenic zebrafish lines are a very reliable means for drug screening.

Therefore, the zebrafish has emerged as a valuable means as an in vivo evaluation platform for screening anti-angiogenesis drugs from chemical compounds [36], microbial extracts [37] and Chinese herbal extracts [38].

Indeed, several transgenic lines of zebrafish have been reported as a useful means to detect environmental toxicants and mutagens [113].

The zebrafish embryo is an excellent means to study whether pre- or post-clinical medicines can affect developing embryos.

Please see lines 41-42 ,61-62, 173-175, 493-494 and 528-529 in the revised manuscript.

Reviewer 2 Report

this is a very nice review of a burgeoning field using zebrafish.

maybe look at the manuscript for any typos but really a great job

Author Response

Reviewer 2

this is a very nice review of a burgeoning field using zebrafish. maybe look at the manuscript for any typos but really a great job

Authors’ response

Thank you.

Reviewer 3 Report

Thank you for the opportunity to review this manuscript. The current article (a review?) provides a simple overview of MANY examples where the zebrafish model has successfully been applied as a means to identify new compounds to treat primarily rare or variant diseases. Now that the comparative genomic analyses between zebrafish and the human
(and most murine genome) counterpart have been performed (circa 2013), this reviewer would expect more. While this screening model is proposed as a replacement to mice, the authors do not address the limitations and
challenges of this model.
1) As an example, this model does not allow for pharmacokinetic (PK) analysis, classical (but regulatory required) pharmacologic analysis, or more traditional formulation determinations. Instead, this model provides a rapid
means for monitoring (and imaging!) physiologic variation assessment or pharmacodynamic (PD) assessments.
2) Gene editing is a big plus in this system and used in most of the examples in this article (being a means to screen compounds for disease variants). However, duplicates of genes exist within the zebrafish model, which
significantly complicates generation of knockout/in strains using either forward or reverse-genetic approaches. This issue, or how researchers have overcome these issues, are not addressed.
3) Very few validated zebrafish reagents (such as antibodies and cell lines) are available to the broad research community. This significantly curtails the in-depth investigation of certain classes of agents, such as molecular and
cellular targeted, or details implicated in a given phenotypes. Thus, it should be discussed the classes/types of agents (or global compound characteristics) that are able to efficiently applied into a high-throughput screen in the zebrafish model.
4) Many of the clinical trials reported finished more than 2 years ago, and several more than5 years ago. Why have there been gaps or no results from these trials? Did the trials end up failing? Have any of these investigational
therapies been successfully applied/approved within their respective populations (especially rare disease)? Has there been an analysis to show that the zebrafish-derived outcomes/compounds are not achieving a greater
percentage of success when translated into patients?
5) Some water-insoluble small molecules cannot be administered to zebrafish because carrier solvents (such as EtOH or DMSO) would reach toxic levels before solubility is achieved. Also, the mechanism of dosing this model is not standard and makes determining ‘dose’ complicated. How is this determined/assessed?
6) To this reviewer's knowledge, zebrafish have never been successfully allometrically scaled to mammals. These further limits the other traditional clinical pharmacology translations necessary in more FDA and EU applications for proof of human success. Translation of zebrafish to mammalian nonclinical models should be discussed.
7) Where are the examples where results from a zebrafish compound screen have been validated and/or verified in a mammalian model? Or what level of evidence has shown a jump from zebrafish to human trials immediately?
8) Considering zebrafish have been considered a 'validated' method since 2013 when the genomic data was finally compared with the human genome, why is it that this platform isn’t used more widely? Adaptation issues? Facility
issues? Limitation in pharmacologic disposition data? Did not translate into mammals? None of this is discussed and only presents a limited view of a 'positive' window. Because of these challenges and limitations listed above, there is a noticeable gap of information in this article as how to take a screened compound and successfully apply it to mammals and then humans. I would also suggest the addition of information into what sort of infrastructure or considerations need to be considered if an institution wished to develop a zebrafish facility. This wort of lacked information would help determine that, despite zebrafish have been successfully employed as a screening model from studies back in 2008, why hasn’t it caught on as a ‘viable’ model within the scientific community when other recent models that have been developed have been added to “main-stream” screening platforms in an average of 4-5 years.

Other minor comments: 
• There appears to be a missing 4th author. Or is it that the “and” should also be a superscript?
• Line 127: add reference.
• Line 130: add reference.
• Line 138: looks like a reference missed your automated citation manager.
• Line 156: Do zebrafish allografts behave the same as the human tumor? Do they have the same pathology too? We know xenografts appear to have the seem pathology.
• Line 187: I would say PD over PK.
• Line 337: there is no figure 1d.

Despite my comments above, this reviewer does share the same overall belief that the zebrafish model does have significance – in a given arena and for certain evaluations. This is why, instead of rejecting this article, I would recommend a major revision and potentially more time than the typical 10 day turn around. I believe a 20-30 day would very much be possible.

Author Response

Reviewer 3

Thank you for the opportunity to review this manuscript. The current article (a review?) provides a simple overview of MANY examples where the zebrafish model has successfully been applied as a means to identify new compounds to treat primarily rare or variant diseases. Now that the comparative genomic analyses between zebrafish and the human (and most murine genome) counterpart have been performed (circa 2013), this reviewer would expect more. While this screening model is proposed as a replacement to mice, the authors do not address the limitations and challenges of this model.

1) As an example, this model does not allow for pharmacokinetic (PK) analysis, classical (but regulatory required) pharmacologic analysis, or more traditional formulation determinations. Instead, this model provides a rapid means for monitoring (and imaging!) physiologic variation assessment or pharmacodynamic (PD) assessments.

Authors’ response to #1:

This article describes how to take advantage of zebrafish as a model to perform preliminary large-scale drug screening. It is not intended to replace the mouse model. As you say, the zebrafish model provides a rapid means for monitoring and assessing physiologic variation or assessing pharmacodynamics. Yet, no dose conversion exists between zebrafish and human, making the translation of pharmacological findings to higher vertebrates very limited without an internal exposure-response relationship (Morgan et al., 2012; Van Wijk et al., 2016). Thus, for pharmacological research, it is necessary to develop a mechanistic understanding of the pharmacokinetic processes in zebrafish larvae in order to quantitatively scale this to higher vertebrates. Kantae et al. (2016) developed a new methodology to characterize drug elimination processes in 3-day-old zebrafish larvae, proving for the first time the feasibility of studies on drug PK and metabolism in zebrafish. However, more evidence is needed. We added these statements in the revised manuscript, including the limitations and challenges of this model. (Please see lines 605-640 in the revised manuscript).

2) Gene editing is a big plus in this system and used in most of the examples in this article (being a means to screen compounds for disease variants). However, duplicates of genes exist within the zebrafish model, which significantly complicates generation of knockout/in strains using either forward or reverse-genetic approaches. This issue, or how researchers have overcome these issues, are not addressed.

Authors’ response to #2:

We agree that duplication of genes in the zebrafish model is one of disadvantages of using zebrafish, as we described in “12. The limitations and challenges of zebrafish model for drug screening” section. However, based on the outcomes published everywhere, it is still possible to successfully generate knockout/in zebrafish lines using either forward or reverse genetics approaches, and the reasons include fecundity (one pair of parents can possibly produce 100 to 200 eggs), light-controlled spawning, large-sized and transparent embryos, short life-cycle and easy manipulation of gene transfer. Therefore, although the success rate is quite low, researchers still have a chance to obtain transgenic zebrafish lines harboring knockout/in at duplicate genes through intensive transgenesis and efficient screening from thousands of microinjected eggs. We added these statements in the revised manuscript (Please see lines 605-640 in the revised manuscript).

3) Very few validated zebrafish reagents (such as antibodies and cell lines) are available to the broad research community. This significantly curtails the in-depth investigation of certain classes of agents, such as molecular and cellular targeted, or details implicated in a given phenotypes. Thus, it should be discussed the classes/types of agents (or global compound characteristics) that are able to efficiently applied into a high-throughput screen in the zebrafish model.

Authors’ response to #3:

Yes, compared to the mouse model, very few validated zebrafish reagents are readily available at the market at the present time. However, commercial mouse antibodies can sometimes be substituted for zebrafish antibodies. If necessary, researchers can take three to four months to prepare zebrafish- specific antibodies. In fact, zebrafish reagents are becoming more and more popular in the marketplace compared to a decade ago. Thus, we believe the availability of zebrafish reagents could quickly turn around because of the increased use of the zebrafish model.

4) Many of the clinical trials reported finished more than 2 years ago, and several more than 5 years ago. Why have there been gaps or no results from these trials? Did the trials end up failing? Have any of these investigational therapies been successfully applied/approved within their respective populations (especially rare disease)? Has there been an analysis to show that the zebrafish-derived outcomes/compounds are not achieving a greater percentage of success when translated into patients?

Authors’ response to #4:

Although interesting, the history of clinical trials in the drug discovery process was not our aim in this article; however, we did note one drug, termed Sunitinib, a VEGF inhibitor, that could inhibit hypoxia-induced invasion, dissemination, and metastasis of T241 tumors in a zebrafish tumor model (Lee et al., 2009) and that has been approved by the FDA.

The high failure rate in drug trials is not necessarily associated with the type of animal models used during the screening process, zebrafish or otherwise. In fact, a 96% failure rate in drug development has been reported, including a 90% failure rate during clinical development (Paul et al., 2010; Pammolli et al., 2011; Scannell et al., 2012; Hay et al., 2014). Such high failure rates result from a new mechanism of action against a previously ‘undrugged’ protein and for diseases where the pathogenesis is still poorly understood.

Cost-cutting in the pharmaceutical industry could be achieved without compromising innovation in R&D practices. For example, culturing zebrafish offers the most advantage for high-throughput drug screening since the maintenance cost of zebrafish is around one tenth that of mice. After preliminary high-throughput rapidly screening, researchers could rule out most negative candidates and only save the most promising candidates for further validation on the mammalian system. This strategy would be cost-effective for finding a new medicine, which is the main concept of this manuscript.

5) Some water-insoluble small molecules cannot be administered to zebrafish because carrier solvents (such as EtOH or DMSO) would reach toxic levels before solubility is achieved. Also, the mechanism of dosing this model is not standard and makes determining ‘dose’ complicated. How is this determined/assessed?

Authors’ response to #5:

Hallare et al. (2006) have demonstrated that less than 1.5% and 1% (v/v) of DMSO and ETOH, respectively, had no effect on zebrafish at the early developmental stage. This finding popularized the use of DMSO or ETOH as a carrier solvent to investigate the effect of water-insoluble molecules on the embryonic development of zebrafish. Taking DMSO as an example, in order to avoid toxicity, researchers employed only 0.1% (v/v) DMSO to immerse zebrafish embryos (McGown et al., 2016; Seda et al., 2019). In fact, the 0.1% (v/v) DMSO is also the common concentration used to screen drugs for cell line or iPSC (Kondo et al., 2017; Chen et al., 2015).

6) To this reviewer's knowledge, zebrafish have never been successfully allometrically scaled to mammals. These further limit the other traditional clinical pharmacology translations necessary in more FDA and EU applications for proof of human success. Translation of zebrafish to mammalian nonclinical models should be discussed.

Authors’ response to #6:

Although the zebrafish system is a well-established vertebrate model, its utilization has been limited owing to the gaps in our understanding of basic pharmacokinetic (PK) processes, such as absorption, distribution, and clearance, that drive the exposure to drugs and drug metabolites. A few studies have reported the expanding allometrical scale from zebrafish to other vertebrates. Some other studies suggest quantitative similarities in metabolite formation between zebrafish larvae and humans (Li et al., 2011; Chng et al., 2012; Hu et al., 2012). Furthermore, Kantae et al. (2016) developed a new methodology to characterize drug elimination processes in 3-day-old zebrafish embryos, proving, for the first time, the feasibility of studies on drug PK and metabolism in zebrafish. Although zebrafish have been used as a model animal for the last two to three decades, more time is still needed to accumulate evidence for this issue. We added this statement in the revised version (Please see lines 605-640 in the revised manuscript).

7) Where are the examples where results from a zebrafish compound screen have been validated and/or verified in a mammalian model? Or what level of evidence has shown a jump from zebrafish to human trials immediately?

Authors’ response to #7:

As we described in the manuscript (please see lines 420-423 and 444-449 in the revised manuscript), many reports have demonstrated that drugs identified from a zebrafish compound screen have also been validated and verified in a mouse model. For example, a small molecule named 16,16-dimethyl- PGE2 (dmPGE2; ProHema) was identified and shown to modulate vertebrate hematopoietic stem cell (HSC) homeostasis in zebrafish (North et al., 2007 [108]). This drug has been validated in a mouse model (Poter et al., 2013; Broxmeyer and Pelus, 2014) which is highly relevant to the goal of enhancing engraftment in human clinical hematopoietic cell transplantation. Macari et al. [109] employed a mutant zebrafish, Rps29-/-, for novel compound screening. They found several calmodulin (CaM) inhibitors that could successfully rescue hemoglobin level in the mutant embryos. Subsequently, they demonstrated that the effect of the CaM inhibitor trifluoperazine (TFP) in human and murine models is also consistent with its effect on zebrafish.

8) Considering zebrafish have been considered a 'validated' method since 2013 when the genomic data was finally compared with the human genome, why is it that this platform isn’t used more widely? Adaptation issues? Facility issues? Limitation in pharmacologic disposition data? Did not translate into mammals? None of this is discussed and only presents a limited view of a 'positive' window. Because of these challenges and limitations listed above, there is a noticeable gap of information in this article as how to take a screened compound and successfully apply it to mammals and then humans. I would also suggest the addition of information into what sort of infrastructure or considerations need to be considered if an institution wished to develop a zebrafish facility. This wort of lacked information would help determine that, despite zebrafish have been successfully employed as a screening model from studies back in 2008, why hasn’t it caught on as a ‘viable’ model within the scientific community when other recent models that have been developed have been added to “main-stream” screening platforms in an average of 4-5 years.

Authors’ response to #8:

Although zebrafish have been successfully employed as a screening model as early as 2008, as you mentioned, many innovations and advanced molecular techniques have since been achieved to make zebrafish a more powerful model organism for high-throughput drug screening in the recent 10 years. With the advent of genome editing to produce genetically modified and mutant transgenic lines of zebrafish, more and more human genetic diseases have been generated in the zebrafish model. For example, as we described in the text, numerous methods allow us to study zebrafish on the basis of their modes of social behavior. Many reports also explore the evolutionary conservation of a subcortical social brain between zebrafish and mammals as the biological basis for using zebrafish to model human social behavior disorders.

A big gap does separate zebrafish and human by the numerous physiological differences, such as blood/plasma composition, host immune response and temperature consistency. On the other hand, we have to appreciate zebrafish as a powerful and economical in vivo animal model for drug screening and assessing the effect of drugs on embryonic development. Before moving on to more laborious pharmacokinetic studies and more costly mammalian models, the zebrafish is a good intermediate alternative in vivo platform to screen for potential drugs and proteins on a large scale. Our article does not intend to suggest the replacement of the mouse model, but rather take advantage of the merits of the zebrafish model as one step of drug screening. For example, we could select candidate drugs which are not toxic and that are target-specific with no side effects and high curability for further study in the mouse model.

Some challenges and limitations you suggested were added in the 12th topic (Please see lines 605-640 in the revised manuscript). However, other issues, such as the limitation in pharmacologic disposition data and slower development into the “main-stream” screening platform, are beyond the scope of this manuscript. As we noted previously, zebrafish have been an animal model for three decades and successfully employed as a screening model since 2008. Thus, compared to the mouse model, the zebrafish model is still in its nascency. Thus, it is true that the zebrafish model is not more widely used for drug screening at the present time. However, if more researchers use zebrafish as an experimental material and more advanced techniques are employed, we believe most questions you proposed might be gradually answered in the future. Our aim in this article was to strengthen the idea that zebrafish could be successfully employed as an alternative intermediate model for large-scale drug screening. We believe that this article will inspire researchers to consider zebrafish in a new light in the course of modern-day drug discovery.

Answers to questions of administration and facilities can be found in materials published by the U.S. Zebrafish International Resource Center (ZIRC) and European Zebrafish Resource Center (EZRC).

Other minor comments: 
• 1. There appears to be a missing 4th author. Or is it that the “and” should also be a superscript?

  • 2. Line 127: add reference. ?
    • 3. Line 130: add reference. ?
    • 4. Line 138: looks like a reference missed your automated citation manager. ?
    • 5. Line 156: Do zebrafish allografts behave the same as the human tumor? Do they have the same pathology too? We know xenografts appear to have the seem pathology.
  • 6. Line 187: I would say PD over PK. ?
    • 7. Line 337: there is no figure 1d.

Authors’ response:

  1. We only have 3 authors.
  2. Thank you We corrected it. Please see line 155 in the revised manuscript.
  3. Thank you We corrected it. Please see line 160 in the revised manuscript.
  4. Thank you We corrected it. Please see line 168 in the revised manuscript.
  5. In zebrafish allograft assays, several methods for donor zebrafish suffering from cancer have been described (Patton et al., 2005; Chen et al., 2007; Sabaawy et al., 2006; Frazer et al., 2009; Mizgirev et al., 2006, 2010; Smith et al., 2010; Tang et al., 2014). Zebrafish recipients require preconditioning treatment if they are not syngeneic or immunosuppressed individuals. Actually, many studies proved that the zebrafish allografts behave the similar property and pathology in comparable with human tumor (Langenau et al., 2003, 2008; Moore et al., 2016; Letrado et al., 2018).
  6. Thank you We corrected it. Please see line 226 in the revised manuscript.
  7. Thank you We corrected it. Please see line 402 in the revised manuscript.

Despite my comments above, this reviewer does share the same overall belief that the zebrafish model does have significance – in a given arena and for certain evaluations. This is why, instead of rejecting this article, I would recommend a major revision and potentially more time than the typical 10 day turn around. I believe a 20-30 day would very much be possible.

Round 2

Reviewer 3 Report

Thank you for allowing me to review your revised manuscript after taking into account my prior comments. 

While my initial comments could be considered a little harsh and out of scope, I think the authors have made a significant effort to address these concerns to help refine the focus and note the potential limitations of the model system that had me concerned. As a clinical pharmacologist, screening platforms are commonly employed, but only useful if we have the translational foresight to make sure the results generated can help us reduce the poor statistics seen in the clinical trials community (as also noted by the authors in their responses). 

This reviewer believes that the revised submission helps clarify many points now to a wider audience compared to the original submisison. No further comments to provide.